# Angiotensin II Induces Automatic Activity of the Isolated Guinea Pig Pulmonary Vein Myocardium through Activation of the IP_3_ Receptor and the Na^+^-Ca^2+^ Exchanger

**DOI:** 10.3390/ijms20071768

**Published:** 2019-04-10

**Authors:** Yusuke Tanaka, Kae Obata, Tamano Ohmori, Kohei Ishiwata, Manato Abe, Shogo Hamaguchi, Iyuki Namekata, Hikaru Tanaka

**Affiliations:** Department of Pharmacology, Faculty of Pharmacological Science, Toho University, Funabashi, Chiba 274-8510, Japan; 3016005t@nc.toho-u.ac.jp (Y.T.); 1013061o@nc.toho-u.ac.jp (K.O.); 1014056o@nc.toho-u.ac.jp (T.O.); 1014024i@nc.toho-u.ac.jp (K.I.); 1014006a@nc.toho-u.ac.jp (M.A.); iyuki@phar.toho-u.ac.jp (I.N.)

**Keywords:** angiotensin II, pulmonary vein myocardium, automaticity

## Abstract

The automaticity of the pulmonary vein myocardium is known to be the major cause of atrial fibrillation. We examined the involvement of angiotensin II in the automatic activity of isolated guinea pig pulmonary vein preparations. In tissue preparations, application of angiotensin II induced an automatic contractile activity; this effect was mimicked by angiotensin I and blocked by losartan, but not by PD123,319 or carvedilol. In cardiomyocytes, application of angiotensin II induced an increase in the frequency of spontaneous Ca^2+^ sparks and the generation of Ca^2+^ transients; these effects were inhibited by losartan or xestospongin C. In tissue preparations, angiotensin II caused membrane potential oscillations, which lead to repetitive generation of action potentials. Angiotensin II increased the diastolic depolarization slope of the spontaneous or evoked action potentials. These effects of angiotensin II were inhibited by SEA0400. In tissue preparations showing spontaneous firing of action potentials, losartan, xestospongin C or SEA0400 decreased the slope of the diastolic depolarization and inhibited the firing of action potentials. In conclusion, in the guinea pig pulmonary vein myocardium, angiotensin II induces the generation of automatic activity through activation of the IP_3_ receptor and the Na^+^-Ca^2+^ exchanger.

## 1. Introduction

The pulmonary vein wall contains a myocardial layer connected to the left atrial myocardium which is capable of generating spontaneous or triggered action potentials [1,2,3]. At the end of the 1990s, it was clinically reported that paroxysmal atrial fibrillation is initiated by trains of rapid discharges from the pulmonary vein [4,5]. Since then, the pulmonary vein myocardium has attracted great attention from researchers as a key player in the generation and maintenance of atrial fibrillation. The electrical activity of the pulmonary vein myocardium has been investigated as a target for pharmacological treatment of atrial fibrillation [6,7,8]. Experiments have been performed in isolated pulmonary vein myocardia from many experimental animal species [9,10,11,12,13], and information on the mechanisms for their automaticity is accumulating.

The pulmonary vein myocardium in general has a lower (less negative) resting membrane potential when compared to atrial myocardium, which reflects the lower density of the inwardly rectifying potassium currents in the pulmonary vein cardiomyocytes [13,14,15]. This allows the generation of diastolic depolarization and the firing of action potentials. Concerning the direct cause of the diastolic depolarization, several sarcolemmal currents were proposed including the Na^+^-Ca^2+^ exchanger current [16,17,18], persistent Na^+^ current [19,20], the Ca^2+^-activated chloride current [21] and the stretch activated current [22]. These membrane currents are dependent on intracellular Ca^2+^ and/or cause transsarcolemmal Ca^2+^ influx. Thus, analysis of intracellular Ca^2+^ movements, as well as of sarcolemmal ion currents, is essential for the understanding of the automaticity of the pulmonary vein myocardium.

The clinical risk factors for atrial fibrillation include hypertension, heart failure and cardiac valve diseases [23]. These conditions are accompanied by changes in the neurohumoral status, including increased activity of the sympathetic nervous system and the rennin-angiotensin system. Angiotensin II, the key component of the renin-angiotensin system, stimulates diverse intracellular signaling cascades and enhances cardiac cellular proliferation and production of extracellular matrix proteins in cardiac fibroblasts, leading to cardiac remodeling [24,25]. Treatment with an angiotensin receptor blocker or an angiotensin-converting enzyme inhibitor was reported to inhibit cardiac remodeling in dogs [26,27,28]. Results of clinical trials appear to indicate that inhibition of the rennin-angiotensin system may prevent the new-onset or recurrence of atrial fibrillation [29,30,31,32,33]. Angiotensin II also has acute hypertensive and antidiuretic effects through induction of vascular contraction and aldosterone release. Concerning the acute effect of angiotensin II on the heart, inotropic and chronotropic effects have been reported [34,35,36,37] which probably reflects the direct effect of angiotensin II on myocardial ion channels [38,39,40]. Concerning the pulmonary vein myocardium, enhancement of electrical activity by angiotensin II has been reported in the rabbit [40], but information concerning the mechanisms for enhancement of the automaticity is limited.

In the present study, we intended to clarify the effect of angiotensin II on the automaticity of the guinea pig pulmonary vein myocardium. We performed contractile force measurements, intracellular Ca^2+^ imaging and action potential recordings. The results indicated that angiotensin II enhances the automaticity of the pulmonary vein myocardium through activation of the IP_3_ receptor and enhancement of the Na^+^-Ca^2+^ exchanger.

## 2. Results

### 2.1. Induction of Automatic Contractile Activity by Angiotensin II

About 34% (62/181) of the isolated pulmonary vein tissue preparations showed spontaneous contractile activity and the rest were quiescent. In the quiescent preparations, angiotensin II, at 100 nM and 1 μM, induced an automatic contractile activity (Figure 1, Table 1). The activity was transient; the frequency of contraction reached a peak within 1 min and gradually decreased towards the termination of repetitive contraction. The induction rate, mean duration, and maximum frequency of contraction were higher under 1 μM angiotensin II than under 100 nM. The induction of contractile activity by 1 μM angiotensin II was inhibited by pre-application of losartan (10 μM), a selective angiotensin AT_1_ receptor blocker [41]. PD123,319 (10 μM), a selective angiotensin AT_2_ receptor blocker [41], and carvedilol (0.1 μM), while a dual blocker of α- and β-adrenoceptors [42] had no effect.

### 2.2. Induction of Automatic Contractile Activity by Angiotensin I

In quiescent isolated pulmonary vein tissue preparations, 1 μM angiotensin I induced an automatic contractile activity in 50% of the preparations, which was transient as was the case with angiotensin II (Figure 2, Table 2). The induction of contractile activity by 1 μM angiotensin I was inhibited by pre-application of losartan (10 μM). Captopril (1 μM), an angiotensin-converting enzyme inhibitor [43], also inhibited the induction of contractile activity. Chymostatin (10 μM), a chymase inhibitor [44], had no effect.

### 2.3. Effect of Angiotensin II on Intracellular Ca^2+^ Dynamics

In 40% (6/15) of the isolated pulmonary vein cardiomyocytes, angiotensin II (1 μM) induced automatic Ca^2+^ transients, rises in intracellular Ca^2+^ concentration throughout the cytoplasm (Figure 3). The generation of the Ca^2+^ transients was preceded by a rise in the frequency of spontaneous Ca^2+^ sparks, local rises in Ca^2+^ concentration in confined areas of about 1 μm in diameter. The induction of Ca^2+^ transients and the rise in the frequency of spontaneous Ca^2+^ sparks by angiotensin II were both inhibited by pre-application of losartan (10 μM) or xestospongin C (3 μM), an inhibitor of the IP_3_ receptor on the sarcoplasmic reticulum (SR) membrane [45].

### 2.4. Induction of Automatic Action Potential Firing and Diastolic Depolarization by Angiotensin II

About 30% (40/131) of the isolated pulmonary vein tissue preparations showed spontaneous firing of action potentials and the rest were quiescent. In quiescent preparations, angiotensin II (1 μM) induced automatic firing of action potentials in 58.3% (7/12) of the preparations (Figure 4A,B, Table 3). The generation of action potentials was accompanied by diastolic depolarization and preceded by oscillation of the membrane potential. The induction of action potential as well as the preceding membrane potential oscillation by angiotensin II was concentration-dependently inhibited by pre-application of the IP_3_ receptor inhibitor, xestospongin C [45], and the Na^+^-Ca^2+^ exchanger inhibitor, SEA0400 {2-[4-[(2,5-difluorophenyl) methoxy] phenoxy]-5-ethoxyaniline; [46]}. Another inhibitor of the IP_3_ receptor, 2-aminoethyl diphenylborinate (2-APB; 2 μM; [47]), also tended to inhibit the action potential firing. Another inhibitor of the Na^+^-Ca^2+^ exchanger, Ni^2+^ (2 mM; [48]), and an inhibitor of the ryanodine receptor, ryanodine (0.1 μM; [49]), also inhibited the action potential firing. In preparations showing spontaneous activity, angiotensin II significantly increased the firing frequency (Figure 4C–E, Table 3).

### 2.5. Enhancement of Diastolic Depolarization by Angiotensin II under Constant Firing Frequency

Field electrical stimulation of the quiescent pulmonary vein tissue preparations resulted in constant generation of action potentials which were characterized by a diastolic depolarization phase similar to those of spontaneous action potentials (Figure 5, Table 3). Angiotensin II (1 μM) significantly increased the diastolic depolarization slope of the action potential. The angiotensin II-induced increase in the diastolic depolarization slope was inhibited by pre-application of losartan (10 μM), xestospongin C (3 μM) or SEA0400 (1 μM).

### 2.6. Effect of Losartan, Xestospongin C and SEA0400 on the Spontaneous Automatic Action Potentials

The spontaneous electrical activity in pulmonary vein myocardium was suppressed by application of losartan (10 μM), xestospongin C (3 μM) or SEA0400 (1 μM) (Figure 6). The frequency of action potential firing, as well as the slope of the diastolic depolarization, was significantly decreased by the application of these agents.

## 3. Discussion

The present study was undertaken to clarify the effect of angiotensin II on the automatic activity of the isolated guinea pig pulmonary vein myocardium, and the following conclusions were obtained. (1) Angiotensin II and angiotensin I induce automatic activity of the myocardium through direct stimulation of the AT_1_ receptor on the cardiomyocytes; (2) AT_1_ receptor stimulation increases the frequency of Ca^2+^ spark firing through activation of the IP_3_ receptor; (3) Angiotensin II enhances the diastolic depolarization by activation of the Na^+^-Ca^2+^ exchanger; (4) The spontaneous electrical activity of the pulmonary vein myocardium is partly mediated by endogenous angiotensin II acting through the same mechanisms as those for exogenous angiotensin II.

We have been studying the basic mechanisms of the automatic activity of the guinea pig pulmonary vein myocardium and reported the involvement of intracellular Ca^2+^ movements and the Na^+^-Ca^2+^ exchanger [13,17]. Such mechanisms are probably involved in the mechanisms of action of various neurohumoral substances on the pulmonary vein myocardium.

In the present study, 100 nM and 1 μM angiotensin II induced a transient automatic activity; the incidence of automatic activity as well as the firing frequency was concentration dependent (Figure 1, Table 1). The transient nature of the induced automatic activity was probably the result of desensitization to angiotensin at the receptor level, which is a common phenomenon observed with G protein coupled receptors [50,51]. The chronotropic and inotropic effects of angiotensin II, as well as the induction of ectopic electrical activity, are reported to be mediated by AT_1_ receptors. AT_2_ receptors are generally reported to have the opposite effects [52]. Concerning the automatic activity of the guinea pig pulmonary vein myocardium, the results with antagonists showed the involvement of AT_1_ but not AT_2_ receptors. AT receptors were shown to be also present on the sympathetic nerve terminals and in some cases they may affect the function of the myocardium including that in the pulmonary vein. [18,53,54,55,56,57,58,59]. In the present study, lack of effect of carvedilol pre-application excluded the involvement of adrenoceptors in the effects of angiotensin II.

Angiotensin I, the precursor of angiotensin II, showed effects similar to those of angiotensin II (Figure 2, Table 2). The existence of the angiotensin-converting enzyme in the vascular endothelial cells and the conversion of angiotensin I to angiotensin II was reported in the pulmonary blood vessels [60,61]. In the present study, results with captopril and chymostatin indicated that the angiotensin-converting enzyme, but not chymase, was involved in the conversion of angiotensin I to angiotensin II in the guinea pig pulmonary vein.

To clarify the effects of angiotensin II on the intracellular Ca^2+^ movements, the pulmonary vein cardiomyocytes were observed with confocal microscopy (Figure 3). Application of angiotensin II induced Ca^2+^ transients, which reflects the generation of action potentials. The induction of action potentials was completely inhibited by losartan, which provides evidence that angiotensin II binds to AT_1_ receptors on the cardiomyocyte. The generation of action potentials was preceded by a rise in the frequency of spontaneous Ca^2+^ sparks, which reflect Ca^2+^ release from the SR. This suggests that intracellular Ca^2+^ released from the SR is involved in the generation of action potentials. In fact, the angiotensin II-induced generation of action potentials, as well as the rise in the frequency of Ca^2+^ sparks, was markedly inhibited by xestospongin C, an inhibitor of the IP_3_ receptor on the SR. Angiotensin II promotes the production of IP_3_, which stimulates the release of Ca^2+^ through IP_3_ receptors on the SR membrane [61,62,63]. Activity of the IP_3_ receptor to release Ca^2+^ was postulated to be involved in the pacemaking of the sino-atrial node [64] and in that of the ectopic pacemakers including the pulmonary vein myocardium [18,65]. In the case of the guinea pig pulmonary vein cardiomyocyte, angiotensin II probably induces automatic activity through enhancement of Ca^2+^ release from the SR through IP_3_ receptors.

The mechanism by which activation of intracellular Ca^2+^ movements by angiotensin II leads to the generation of electrical activity was examined by microelectrode experiments. In quiescent tissue preparations, angiotensin II induced a repetitive firing of action potentials (Figure 4A,B, Table 3). This was often preceded by an oscillation of the resting membrane potential, which probably reflects an oscillation of intracellular Ca^2+^ concentration, as was the case with ouabain-induced automatic activity [17]. The angiotensin II-induced action potentials had a diastolic depolarization phase characteristic of myocardia with automaticity. In preparations showing spontaneous firing of action potentials, angiotensin II increased the slope of the diastolic depolarization and the frequency of action potential firing (Figure 4C–E; Table 3). The effect on other action potential parameters was minimum, indicating that the angiotensin II-induced increase in firing frequency is mostly caused by the increase in the diastolic depolarization slope. Increase in the depolarization slope by angiotensin II was also observed in preparations firing at a constant frequency of 1 Hz under field stimulation (Figure 5), which suggests that the enhancement of diastolic depolarization by angiotensin II is the cause rather than the result of the increase in firing frequency.

The increase in diastolic depolarization slope (Figure 5B,C), as well as the induction of spontaneous firing (Figure 4B), by angiotensin II was inhibited by inhibitors of the Na^+^-Ca^2+^ exchanger. The forward mode of the Na^+^-Ca^2+^ exchanger causes depolarization when it pumps out 3 intracellular Na^+^ in exchange for one intracellular Ca^2+^. Inhibition of the diastolic depolarization of the pulmonary vein myocardium by SEA0400 was also observed with ouabain- and tertiapin-induced automatic activity in the guinea pig pulmonary vein myocardium [13,17]. Inhibition of automatic activity with inhibitors of the Na^+^-Ca^2+^ exchanger was also observed in the rabbit [16] and rat [18] pulmonary vein myocardium. In the rabbit pulmonary vein cardiomyocyte, angiotensin II was reported to increase the Na^+^-Ca^2+^ exchanger current [40]. Thus, the Na^+^-Ca^2+^ exchanger appears to be a major depolarizing mechanism in the pulmonary vein myocardium. The angiotensin II-induced generation of spontaneous action potentials (Figure 4B) and the increase in the diastolic depolarization slope (Figure 5B) were inhibited by inhibitors of the IP_3_ receptor indicating that a significant fraction of the Ca^2+^ supplied to the Na^+^-Ca^2+^ exchanger is provided by Ca^2+^ released from the SR through the IP_3_ receptor. This is similar to the case with noradrenaline-induced automatic activity of the rat pulmonary vein myocardium, in which inhibition was observed with 2-APB [18]. Inhibition of the angiotensin II-induced generation of action potential firing by ryanodine (Figure 4B) indicates the functional presence of ryanodine receptors in the pulmonary vein myocardium. As the mechanism of action of ryanodine is complex [49], the precise mechanisms of Ca^2+^ release from the SR in the pulmonary vein myocardium awaits further investigation.

Spontaneous activity, in the absence of exogenously applied angiotensin II, was observed in about 30% of the guinea pig isolated pulmonary vein tissue preparations [17]. The spontaneous firing of the action potential, as well as the slope of the diastolic depolarization, was partially but significantly inhibited by losartan (Figure 6A), indicating that endogenous angiotensin II is involved in the generation of spontaneous activity. The observation that mechanical stretch applied to cultured cardiomyocytes induces the secretion of stored angiotensin II into the culture medium implies an autocrine mechanism [66]. Partial inhibition of spontaneous activity and the diastolic depolarization slope was also observed with xestospongin C and SEA0400 (Figure 6B,C). This indicates that the spontaneous electrical activity involves the IP_3_ receptor and the Na^+^-Ca^2+^ exchanger, mechanisms in common with the automatic activity induced by exogenous angiotensin II. At the same time, it suggests that other mechanisms also contribute to the automatic activity. Concerning the membrane currents underlying the diastolic depolarization of the pulmonary vein myocardium, the involvement of the persistent Na^+^ current [19,20], the Ca^2+^-activated chloride current [21] and the stretch activated current [22] has been postulated. In the rabbit pulmonary vein cardiomyocyte, alterations in various membrane currents including the L-type Ca^2+^ current and the potassium currents were observed [40]. Thus, it is possible that angiotensin II affects the automaticity of the pulmonary vein myocardium through multiple mechanisms.

The present results suggested that the acute effect of angiotensin II is partly responsible for the automatic activity of the pulmonary vein myocardium. This implies that application of angiotensin II receptor blockers or angiotensin-converting enzyme inhibitors may be effective against atrial fibrillation. Results of clinical trials appear to indicate that inhibition of the rennin-angiotensin system may prevent the new-onset or recurrence of atrial fibrillation, but negative results were also reported and a consensus has not yet been reached [29,30,31,32,33,67]. Such variation in results may arise from difference in patient selection, duration of the study and endpoints. The pathogenetic importance of the rennin-angiotensin system and/or the automaticity of the pulmonary vein may vary among patients. The automaticity of the pulmonary vein myocardium was reported to be affected by acute and chronic mechanical stretch [22,68] and various neurohumoral factors such as noradrenaline [18,58,59], acetylcholine [15,69], endothelin-1 [70] and nitric oxide [71], which may either augment or attenuate the effect of angiotensin II. Factors related to atrial fibrillation such as mechanical stretch [72] and heart failure [73], as well as atrial fibrillation itself [74], were reported to affect the expression level and function of angiotensin II receptors in the working myocardium. Further investigation of the role of angiotensin II in the pulmonary vein automaticity and pathogenesis of atrial fibrillation would provide a basis for an effective therapeutic strategy against atrial fibrillation.

## 4. Materials and Methods

### 4.1. General

All experiments were performed in compliance with the Guiding Principles for the Care and Use of Laboratory Animals approved by The Japanese Pharmacological Society and the Guide for the Care and Use of Laboratory Animals at Faculty of Pharmaceutical Sciences, Toho University (18-52-362, 7 May 2018). The experimental procedures were basically the same as those in our previous studies [13,17,22].

### 4.2. Preparation of Guinea Pig Pulmonary Vein

Hartley strain male guinea pigs weighing 300 to 450 g were used. The hearts with lungs were quickly removed and tissue preparations were made from the four major pulmonary-vein trunks. They were placed in a 20 mL organ bath containing the physiological salt solution following composition (mM): NaCl 118.4, KCl 4.7, CaCl_2_ 2.5, MgSO_4_ 1.2, KH_2_PO_4_ 1.2, NaHCO_3_ 24.9 and glucose 11.0 (pH 7.4), gassed with 95% O_2_-5% CO_2_ and maintained at 36 ± 0.5 °C. The region of the pulmonary vein close to the orifice was cut open and used for the measurement of contraction and action potential firing.

### 4.3. Contractile Force Measurement

The isometric contractile force of isolated tissue preparations was recorded. One end of the pulmonary vein preparation was pinned down on a silicon block at the bottom of the organ bath and the other end was attached to a needle connected to a force-displacement transducer (TB-611T, Nihon Kohden, Tokyo, Japan). The detected contractile force was amplified with a carrier amplifier (AP-621G, Nihon Kohden, Tokyo) and digitized by an A/D converting interface (Power Lab, AD Instruments, Dunedin, New Zealand).

### 4.4. Action Potential Measurement

Spontaneous and evoked action potentials of tissue preparations were recorded with glass microelectrodes filled with 3 M KCl. In the case of quiescent preparations, rectangular current pulses (1 Hz, 3 msec, 1.5 × threshold voltage) were applied through a pair of platinum plate electrodes generated from an electronic stimulator (SEN-3301, Nihon Kohden) to evoke the action potentials. The action potential parameters: frequency, maximum diastolic potential (MDP); maximum rate of rise (dV/dt)_max_; action potential duration at 90% repolarization (APD_90_) and the slope of the diastolic depolarization phase were measured.

### 4.5. Isolation of Pulmonary Vein Cardiomyocytes and Confocal Microscopy

Hearts with lungs were quickly removed from male Hartley guinea pigs (weight, 350–450 g). After Langendorff perfusion of the heart with the pulmonary veins attached and treatment with 0.5 mg/mL collagenase (YK-102, Yakult, Tokyo, Japan) and 0.1 mg/mL protease (type XIV; Sigma-Aldrich, St. Louis, MO, USA) for about 20 min, the pulmonary vein cardiomyocytes were isolated. The extracellular solution was of the following composition (mM concentration): NaCl 143, KCl 5.4, CaCl_2_ 1.8, MgCl_2_ 1.0, NaH_2_PO_4_ 0.33, glucose 5.5, and *N*-(2-hydroxyethyl) piperazine-*N*-2-ethanesulfonic acid (HEPES) 5.0 (pH 7.4), and gassed with 100% O_2_ at 36.0 ± 0.5 °C.

For the analysis of intracellular Ca^2+^ movements, the cardiomyocytes were treated with 5 μM fluo-4/AM (Dojindo, Kumamoto, Japan) and superfused with the extracellular solution mentioned above at room temperature. The cells were observed with a rapid scanning confocal microscope A1R (Nikon, Tokyo, Japan). The objective lens was Apochromat ×40, 1.15 numerical aperture (water immersion). The excitation wavelength was 488 nm and the emission in the wavelength range of 500 to 550 nm was detected and analyzed at 8.7 msec intervals. The fluorescent intensity at each time point was normalized against the basal intensity.

### 4.6. Chemicals

Angiotensin II (Peptide Institute, Osaka, Japan), losartan (Wako Pure Chemical Industries, Osaka, Japan), PD123,319 (Sigma-Aldrich, St. Louis, MO, USA), angiotensin I (Peptide Institute, Osaka, Japan), captopril (Sigma-Aldrich, St. Louis, MO, USA), chymostatin (Sigma-Aldrich, St. Louis, MO, USA) and nickel(II) chloride hexahydrate (Sigma-Aldrich, St. Louis, MO, USA) were dissolved in water. Carvedilol (Tokyo Chemical Industry, Tokyo, Japan), xestospongin C (Wako Pure Chemical Industries, Osaka, Japan), SEA0400 (synthesized in our faculty according to the reported method [75]), 2-APB (Sigma-Aldrich, St. Louis, MO, USA) and ryanodine (Wako Pure Chemical Industries, Osaka, Japan) were dissolved in dimethyl sulfoxide (DMSO). They were added to the organ bath to obtain the desired final concentration.

### 4.7. Data Analysis and Statistics

Contractile force and action potential parameters were analyzed by Chart 7 (AD instruments, Dunedin, New Zealand) and the Ca^2+^ movements were analyzed by NIS Elements software (Nikon, Tokyo, Japan). Data were expressed as means ± standard error of the mean (S.E.M.). Statistical significance between means was evaluated by the Dunnett’s test for multiple comparisons or Fisher’s exact test, using the GraphPad PRISM 6.07 software (GraphPad Software, San Diego, CA, USA). A *P* value less than 0.05 was considered significant.

## Figures and Tables

**Figure 1 ijms-20-01768-f001:**
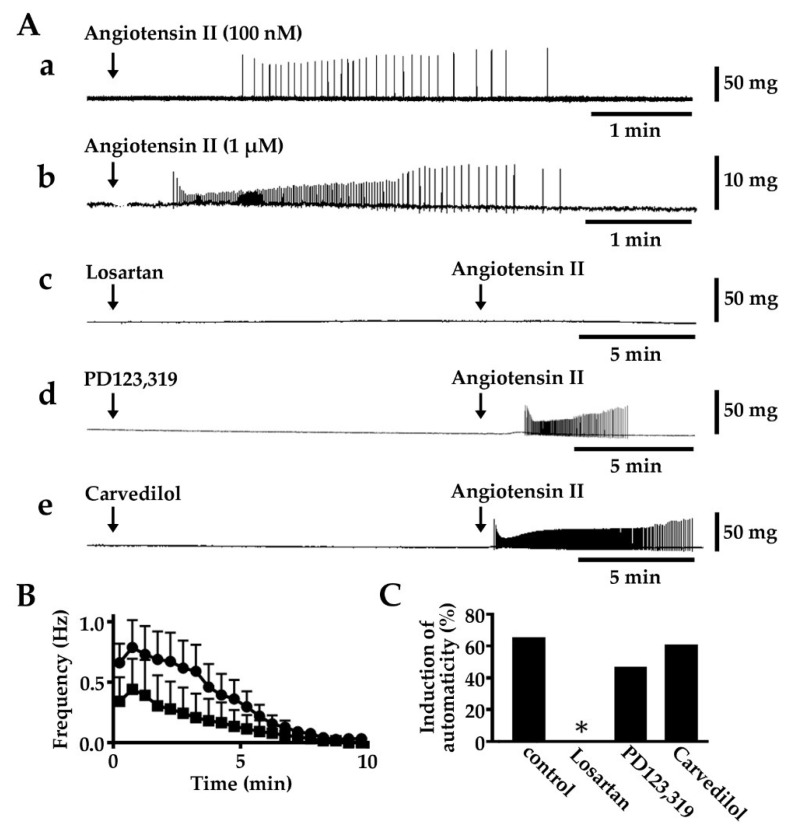
Induction of automatic contractile activity by angiotensin II in pulmonary vein tissue preparations. (**A**) Typical contractile records on application of 100 nM angiotensin II (a), 1 μM angiotensin II (b), 1 μM angiotensin II in the presence of 10 μM losartan (c), 1 μM angiotensin II in the presence of 10 μM PD123,319 (d) and 1 μM angiotensin II in the presence of 100 nM carvedilol (e); (**B**) Summarized time course of the frequency of contraction induced by 100 nM angiotensin II (squares), and 1 μM angiotensin II (circles). The 0 min on the time scale indicates the onset of contractile activity. Symbols and vertical bars indicate the mean ± S.E.M. from 4 and 11 experiments, respectively; (**C**) Summarized results of the rate of induction of automatic contractions. Columns indicate the rate of induction 10 to 17 experiments. Asterisks indicate significant difference from the control (*p* < 0.05) as evaluated by the Fisher’s exact test.

**Figure 2 ijms-20-01768-f002:**
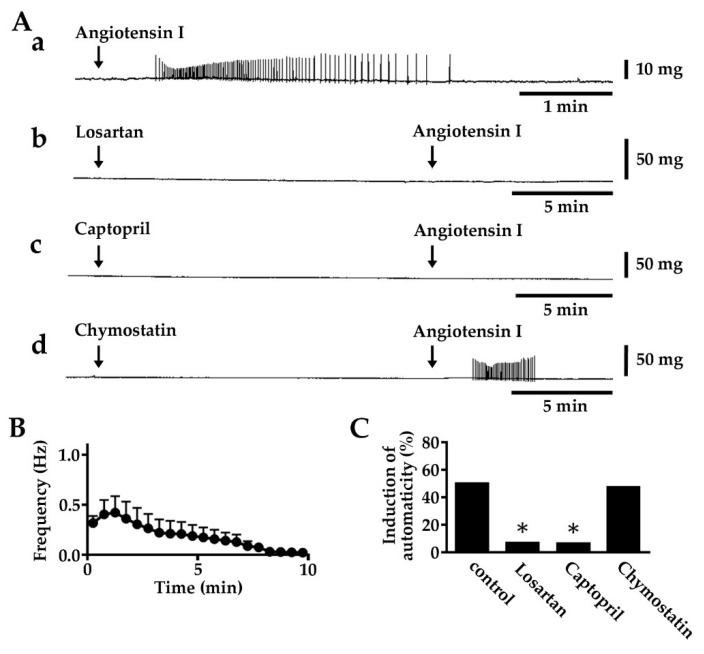
Induction of automatic contractile activity by angiotensin I in pulmonary vein tissue preparations. (**A**) Typical contractile records on application of 1 μM angiotensin I alone (a), in the presence of 10 μM losartan (b), in the presence of 1 μM captopril (c), in the presence of 10 μM chymostatin (d); (**B**) Summarized time course of the frequency of contraction induced by 1 μM angiotensin I. The 0 min on the time scale indicates the onset of contractile activity. Symbols and vertical bars indicate the mean ± S.E.M. from 8 experiments; (**C**) Summarized results of the rate of induction of automatic contractions by angiotensin I. Columns indicate the rate of induction from 15 to 20 experiments. Asterisks indicate significant difference from the control (*p* < 0.05) as evaluated by the Fisher’s exact test.

**Figure 3 ijms-20-01768-f003:**
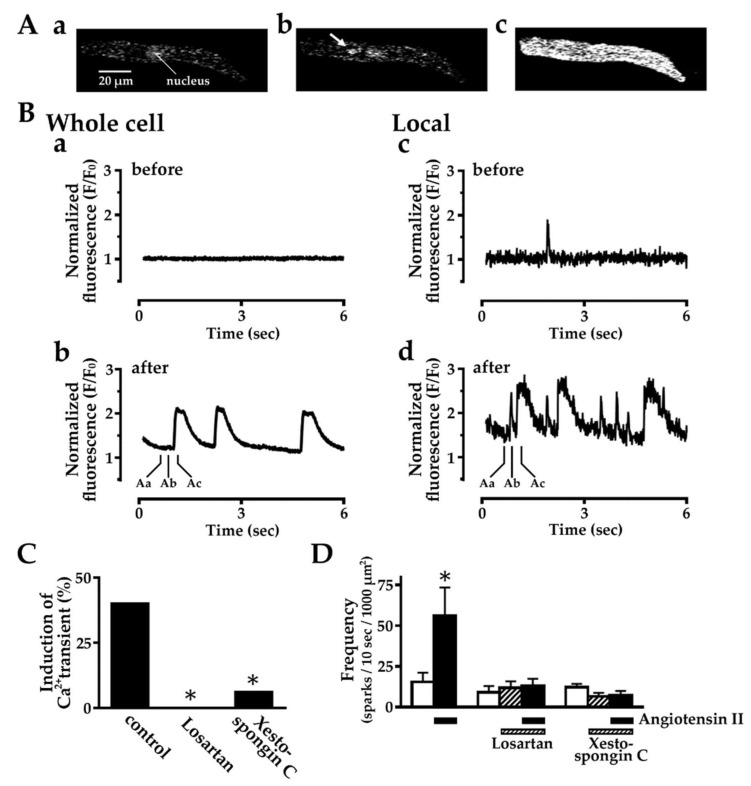
Effect of angiotensin II on intracellular Ca^2+^ dynamics in pulmonary vein cardiomyocytes loaded with the Ca^2+^ sensitive fluoroprobe fluo-4. (**A**) Typical fluorescence images under quiescence (a), on firing of Ca^2+^ sparks (b; arrow) and on firing of a Ca^2+^ transient (c); (**B**) Time courses of the fluorescence of the entire cell (a, b) and in the circular region of 1 μm in diameter at the site shown by the arrow in Ab (c, d) before (a, c) and after the application of 1 μM angiotensin II (b, d). The time points Aa, Ab and Ac in panel b and d corresponds to the panels a, b and c of A; (**C**) Summarized results for the rate of induction of Ca^2+^ transients by angiotensin II. Columns indicate the rate of induction from 14 to 17 experiments. Asterisks indicate statistical significance (*p* < 0.05) from the control as evaluated by the Fisher’s exact test; (**D**) Summarized results for the increase in the frequency of Ca^2+^ spark firing by angiotensin II. Open columns, hatched columns and closed columns indicate the frequency in the absence of agents, after the application of 10 μM losartan or 3 μM xestospongin C, and after further application of 1 μM angiotensin II, respectively. Columns and vertical bars indicate the mean ± S.E.M. from 6 experiments. Asterisks indicate statistical significance (*p* < 0.05) from the value in the absence of agents as evaluated by the Dunnett’s test for multiple comparisons.

**Figure 4 ijms-20-01768-f004:**
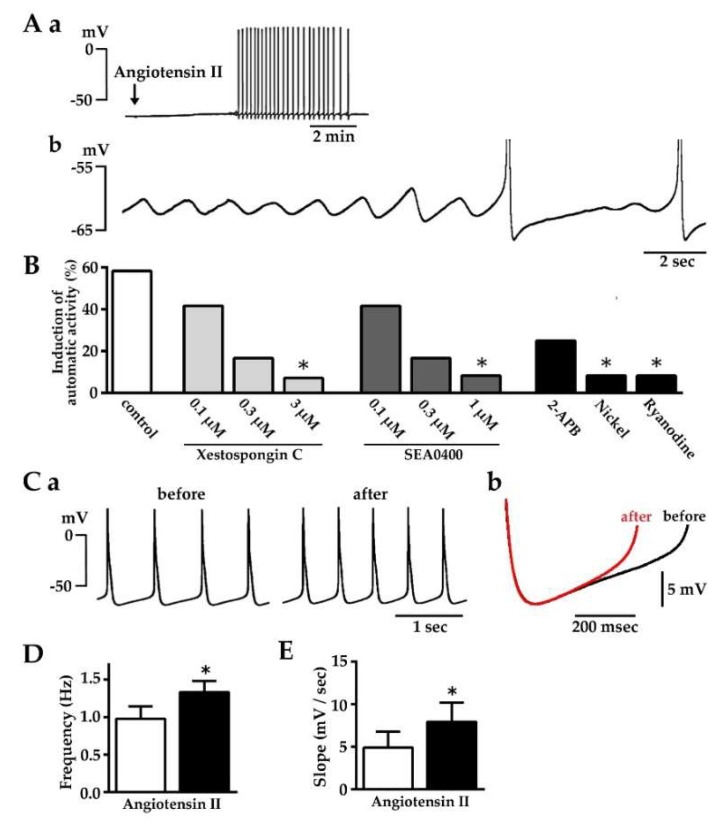
Effects of angiotensin II on automatic action potential firing and diastolic depolarization. (**A**) A typical membrane potential trace on application of 1 μM angiotensin II (a) and an expanded trace after the application showing the oscillation of the membrane potential and diastolic depolarization preceding the action potentials (b); (**B**) Summarized results for the rate of induction of action potentials by 1 μM angiotensin II alone and in the presence of 0.1–3 μM xestospongin C, 0.1-1 μM SEA0400, 2 μM 2-APB, 2 mM Ni^2+^, or 0.1 μM ryanodine. Columns indicate the rate of induction from 12 to 14 experiments. Asterisks indicate statistical significance (*p* < 0.05) from the control as evaluated by the Fisher’s exact test; (**C**) Typical spontaneous action potential traces before and after the application of 1 μM angiotensin II (a) and an overlay of the diastolic depolarization phase before (black) and after (red) the application (b); (**D**) Summarized results for the increase in action potential firing frequency by angiotensin II; (**E**) Summarized results for the increase in the diastolic depolarization slope by angiotensin II. Open and closed columns in **D** and **E** indicate the values before and after the addition of agents, respectively. Columns and vertical bars indicate the mean ± S.E.M. from 6 experiments. Asterisks indicate statistical significance (*p* < 0.05) from the control as evaluated by the paired *t*-test.

**Figure 5 ijms-20-01768-f005:**
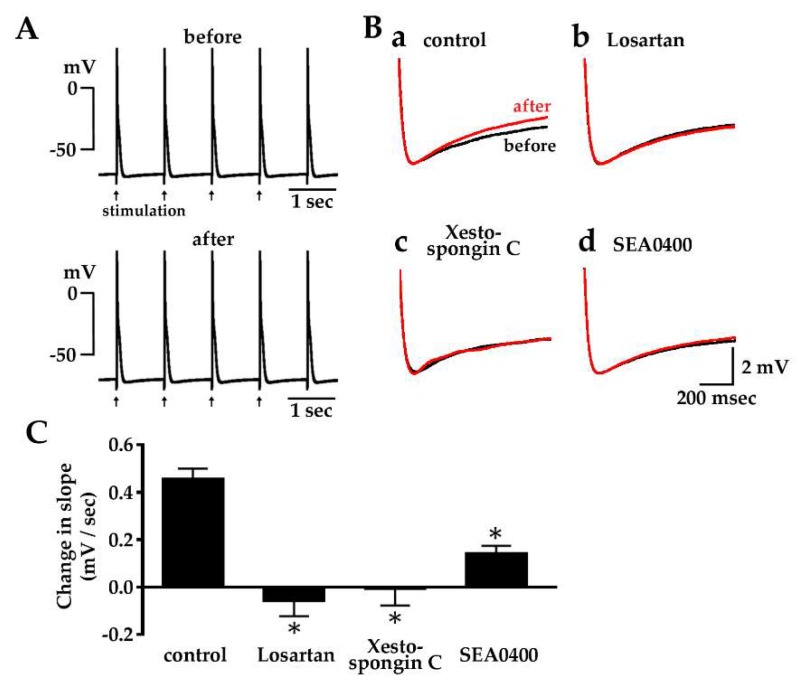
Enhancement of diastolic depolarization by angiotensin II under constant firing frequency. (**A**): Typical action potential traces under 1 Hz stimulation in the absence and presence of 1 μM angiotensin II. Arrows indicate field stimulation at 1 Hz; (**B**) Typical traces of the diastolic depolarization phase before (black) and after (red) the application of 1 μM angiotensin II overlaid for comparison. Angiotensin II was added alone (a) or in the presence of 10 μM losartan (b), 3 μM xestospongin C (c) or 1 μM SEA0400 (d); (**C**) Summarized results for the angiotensin II-induced changes in the diastolic depolarization slope. Columns and vertical bars indicate the mean ± S.E.M. from 6 experiments. Asterisks indicate statistical significance (*p* < 0.05) from the control as evaluated by the Dunnett’s test for multiple comparisons.

**Figure 6 ijms-20-01768-f006:**
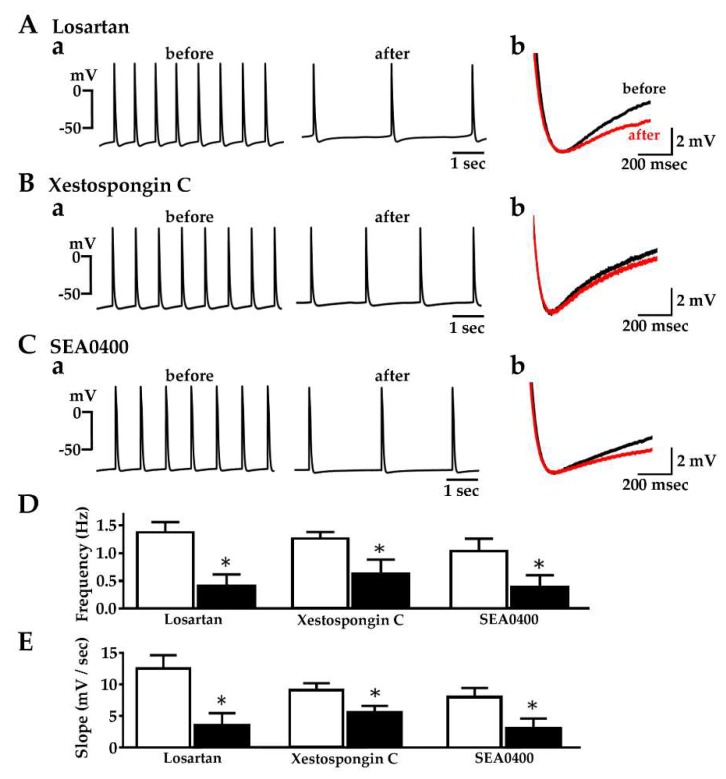
Effect of losartan, xestospongin C and SEA0400 on spontaneous action potentials. (**A**): Typical traces before and after the application of 10 μM losartan (a) and their diastolic depolarization phase overlaid for comparison (b); (**B**) Typical traces before and after the application of 3 μM xestospongin C (a) and their diastolic depolarization phase overlaid for comparison (b); (**C**) Typical traces before and after the application of 1 μM SEA0400 (a) and their diastolic depolarization phase overlaid for comparison (b); (**D**) Summarized results for the decrease in action potential firing frequency; (**E**) Summarized results for the decrease in the diastolic depolarization slope. Open and closed columns in **D** and **E** indicate the values before and after the addition of agents, respectively. Columns and vertical bars indicate the mean ± S.E.M. from 6 experiments. Asterisks indicate statistical significance (*p* < 0.05) as evaluated by the paired *t*-test.

**Table 1 ijms-20-01768-t001:** Induction of automatic contractile activity by angiotensin II.

	Induction of Automatic Activity	Parameters of Induced Automatic Activity			
		Latency (sec)	Duration (min)	Maximum Frequency (Hz)	*n*
Angiotensin II (100 nM)	4/15 (26.7%)	66.4 ± 23.1	4.0 ± 1.6	0.4 ± 0.3	4
Angiotensin II (1 μM)	11/17 (64.7%)	50.4 ± 7.7	4.4 ± 1.1	0.8 ± 0.2	11
Losartan + Angiotensin II (1 μM)	0/10 (0%)	-	-	-	0
PD123,319 + Angiotensin II (1 μM)	6/13 (46.2%)	59.6 ± 17.8	3.5 ± 0.4	0.6 ± 0.1	6
Carvedilol + Angiotensin II (1 μM)	6/10 (60.0%)	33.1 ± 7.9	6.3 ± 1.0	0.6 ± 0.1	6

Angiotensin II was added in the absence and presence of 10 μM losartan, 10 μM PD123,319 or 100 nM carvedilol. The rate of induction of contractile activity and the parameters for the induced activity were indicated. The values are the mean ± S.E.M.

**Table 2 ijms-20-01768-t002:** Induction of automatic contractile activity by angiotensin I.

	Induction of Automatic Activity	Parameters of Induced Automatic Activity			
		Latency (sec)	Duration (min)	Maximum Frequency (Hz)	*n*
Angiotensin I	10/20 (50.0%)	47.8 ± 9.2	9.2 ± 3.1	0.4 ± 0.2	10
Losartan + Angiotensin I	1/15 (6.7%)	7.6	6.1	1.3	1
Captopril + Angiotensin I	1/16 (6.3%)	79.6	2.7	0.3	1
Chymostatin + Angiotensin I	9/19 (47.4%)	62.2 ± 16.1	2.6 ± 0.2	0.4 ± 0.1	9

Angiotensin I (1 μM) was added in the absence and presence of 10 μM losartan, 1 μM captopril or 10 μM chymostatin. The rate of induction of contractile activity and the parameters for the induced activity were indicated. The values are the mean ± S.E.M.

**Table 3 ijms-20-01768-t003:** Effect of angiotensin II on action potential parameters.

	Angiotensin II-Induced Activity (*n* = 7)	Spontaneous Activity (*n* = 6)		Stimulation-Induced Activity (*n* = 6)	
		Before	After	Before	After
Frequency (Hz)	0.51 ± 0.14	0.98 ± 0.02	1.33 ± 0.11 *	-	-
MDP (mV)	−71.0 ± 1.0	−70.4 ± 2.4	−71.5 ± 2.0	−75.6 ± 1.5	−74.7 ± 1.2
Slope (mV/sec)	2.70 ± 0.40	4.90 ± 1.79	7.94 ± 2.12 *	2.66 ± 0.37	3.12 ± 0.38 *
(dV/dt)_max_ (V/sec)	99.3 ± 24.5	116.2 ± 18.2	122.6 ± 16.5	150.7 ± 7.8	148.3 ± 7.8
Amplitude (mV)	92.5 ± 2.5	97.3 ± 3.3	100.7 ± 3.1 *	106.7 ± 1.1	107.5 ± 1.4
APD_90_ (msec)	84.2 ± 2.6	82.3 ± 3.8	85.2 ± 3.3	84.6 ± 2.7	90.7 ± 2.3 *

Action potential parameters before and after the application of angiotensin II (1 μM). The parameters measured were frequency, maximum diastolic potential (MDP), slope of the diastolic depolarization (Slope), maximum rate of rise ((dV/dt)_max_), amplitude and action potential duration at 90% repolarization (APD_90_). The time of parameter measurement after the application of angiotensin II was 1 min for the angiotensin II-induced activity and 3 min for the spontaneous and stimulation-induced activities. The values are the mean ± S.E.M. Asterisks indicate significant difference from the corresponding values before application of angiotensin II as evaluated by the paired *t*-test.

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
