# Peer review of "Angiotensin II Induces Automatic Activity of the Isolated Guinea Pig Pulmonary Vein Myocardium through Activation of the IP3 Receptor and the Na+-Ca2+ Exchanger"

_ijms, 2019, doi:10.3390/ijms20071768_

Round 1
Reviewer 1 Report
In this manuscript, Tanaka et al. examined the effects of AngII on contractile activities, intracellular Ca2+signals, and membrane potential of pulmonary vein myocardium isolated from guinea pigs. Pharmacological inhibitors of AT1R, ACE, IP3R, and NCX were used to dissect the involvement of these molecular structures in the observed effects. Overall, these studies indicate interesting mechanisms whereby the IP3R and NCX are involved in AT1R-mediated effects on contraction and electrical activities of the pulmonary vein myocardium. While the findings are potentially interesting, only single doses were used for all pharmacological inhibitors. Since important mechanistic conclusions regarding the roles of the IP3 receptors and the NCX in AngII-induced Ca2+ transients and diastolic depolarization were made based solely on pharmacological experiments, dose-dependent responses need to be demonstrated.
Author Response
Response to Reviewer 1 Comments
Point: While the findings are potentially interesting, only single doses were used for all pharmacological inhibitors. Since important mechanistic conclusions regarding the roles of the IP3 receptors and the NCX in AngII-induced Ca2+ transients and diastolic depolarization were made based solely on pharmacological experiments, dose-dependent responses need to be demonstrated.
Response: We agree with the reviewer that the pharmacological agents are of crucial importance in this study. To strengthen the pharmacological evidence for the involvement of IP3 receptor and NCX in the induction of activity by angiotensin II, we performed additional experiments with additional inhibitors for IP3 receptor and NCX, which are 2-aminoethyl diphenylborinate and Ni2+, respectively. The results that the induction of activity by angiotensin II was markedly inhibited by these inhibitors confirmed that IP3 receptor and NCX were involved. This was added to the revised manuscript (page 7, line 154-159).
The concentration of the inhibitors used in the original and additional experiments were chosen to maximize the desired inhibitory effect on the target with minimum off-target effects. Thus, we focused on the experiments with these optimum concentrations. The background information concerning the profiles of the agents are available in the references.

Reviewer 2 Report
In this well written MS, authors have elegantly conducted experiments to determine the role various calcium channels activating the And II mediated automaticity of pulmonary vein myocardium. Authors have used various small molecules and inhibitors to decipher the mechanism, which is the strength of the MS. Here are few comments -
What is the rationale for the dose of Ang II, Losartan, PD and Carvedilol used in this study? Did authors do any preliminary study? If authors have followed any previous papers, they should cite the references for all of these small molecules.
Some minor spelling mistakes found in the MS
The discussion is little descriptive, which could be shortened.
Ryanodine receptors are expressed in the pulmonary myocardium. Authors should discuss the same in their discussion.
Did authors investigate the role of ryanodine receptors in the contractile force? Inclusion of this investigation will demonstrate the comprehensive participation of calcium channels in executing the contractile force. Authors should considering a pilot study in presence of ryanodine receptor inhibitors, which would strengthen the MS, further!
Author Response
Response to Reviewer 2 Comments
Point 1: What is the rationale for the dose of Ang II, Losartan, PD and Carvedilol used in this study? Did authors do any preliminary study? If authors have followed any previous papers, they should cite the references for all of these small molecules.
Response 1: The concentration of the inhibitors used in the original and additional experiments were chosen to maximize the desired inhibitory effect on the target with minimum off-target effects. Thus, we focused on the experiments with these optimum concentrations. The background information concerning the profiles of the agents are available in the references.
Point 2: Some minor spelling mistakes found in the MS
Response 2: The entire text was carefully checked for misspellings.
Point 3: The discussion is little descriptive, which could be shortened.
Response 3: The background information on the guinea pig myocardium in general and the description of sympathetic nerves in the pulmonary vein was shortened. As a result, the total discussion was slightly shortened despite the addition of new sentences.
Point 4: Ryanodine receptors are expressed in the pulmonary myocardium. Authors should discuss the same in their discussion.
Response 4: The discussion concerning the presence of the ryanodine receptor in the pulmonary vein cardiomyocyte based on our additional experiment (described below) was added to the discussion (page 12, line 299-303).
Point 5: Did authors investigate the role of ryanodine receptors in the contractile force? Inclusion of this investigation will demonstrate the comprehensive participation of calcium channels in executing the contractile force. Authors should considering a pilot study in presence of ryanodine receptor inhibitors, which would strengthen the MS, further!
Response 5: We performed additional experiments showing that ryanodine inhibits the induction of activity by angiotensin II. This implies that the ryanodine receptor may possibly be in involved in angiotensin II-induced Ca2+ release. However, the mode of action of ryanodine on the ryanodine receptor is complex. It fixes the channel in a semi-open state and eventually causes depletion of the sarcoplasmic reticulum Ca2+. This may result in an inhibition of Ca2+ release from the IP3 receptor. Thus, a definitive conclusion awaits the development of novel agents with specific inhibitory action on the ryanodine receptor.
The results of this additional experiment and a minimum amount of discussion were added to the revised text (page 7, line 154-159 and page 12, line 299-303).

Round 2
Reviewer 1 Report
In this revision, the authors have added a couple of experiments using 2-APB as an additional inhibitor of IP3 receptor and Ni2+ as an inhibitor of the NCX.
Dose-dependency is essential for conclusions of pharmacological studies. The fundamental concern I have with this work is that all experiments were done with a single dose of each pharmacological agent. This questions about the attribution of the observed effects to the supposed targets of the drugs. Instead of generating dose response curves for the effects of any pharmacological agent, the authors chose to test the effects of additional agents, again with single dose for each. The initial concern thus remains. As a specific example, the authors used 2-APB as an additional inhibitor of the IP3 receptor at 2 microM. Figure 4B shows that there was no significant (n.s.) effect of 2-APB at this dose on the automatic activity induced by AngII. However, the authors stated that this agent inhibited both AP firing and preceding membrane potential oscillation induced by AngII (p. 7, lines 154-159). This statement is thus incorrect. Moreover, I do not understand the reason for the choice of this dose (2 microM) for 2-APB; while reference 46 reported that the IC50 value for 2-APB for IP3-induced Ca2+ release is 50 microM. This is not to mention that IC50 values certainly can change depending on the tissue under study. Adding data on another pharmacological agent at a dose that does not generate a significant effect does not add any strength to the initial data for the IP3 receptor.
Overall, as I mentioned in my initial review of the manuscript, the findings are potentially very interesting. However, since the studies only utilized pharmacological inhibitors and no other molecular tools are employed, it is essential that dose-dependent effects be demonstrated to substantiate the conclusions.
Author Response
Response to Reviewer 1 Comments
Point 1: Dose-dependency is essential for conclusions of pharmacological studies. The fundamental concern I have with this work is that all experiments were done with a single dose of each pharmacological agent.
Response 1: In accordance with the reviewer’s suggestion, we examined the effects of different concentrations of xestospongin C and SEA0400 on the induction of action potentials by angiotensin II. The results showed that the effects of both agents were concentration-dependent, which strengthens our conclusions on the mechanisms of action of angiotensin II. The data were presented in Figure 4B.
Point 2: The authors stated that 2-APB inhibited both AP firing and preceding membrane potential oscillation induced by Ang II (p.7, lines 154-159). This statement is thus incorrect.
Response 2: We corrected the statement concerning the effect of 2-APB (p.7, lines 154-161).
Point 3: Moreover, I do not understand the reason for the choice of this dose (2 microM) for 2-APB; while reference 46 reported that the IC50 value for 2-APB for IP3-induced Ca2+ release is 50 microM.
Response 3: We replaced the paper by Maruyama et al. (reference 46 of the first revision) with that by Mackenzie et al. (reference 47 of the second revision). The Maruyama paper is one of the early works with 2-APB but uses microsomal preparations. The Mackenzie paper showed that 2 microM 2-APB, which does not affect normal Ca2+ signaling, inhibits the Ca2+ release from IP3 receptors in cardiomyocytes. Okamoto et al. (reference 18) also uses 2 microM 2-APB for the inhibition of IP3 receptors in cardiomyocytes.
Reviewer 2 Report
Authors have appropriately answered the raised concerns and there are no more suggestions!
Author Response
Response to Reviewer 2 Comments
Point 1: Authors have appropriately answered the raised concerns and there are no more suggestions!
Response 1: Thank you for your review and suggestions.
Round 3
Reviewer 1 Report
The manuscript has been improved. I recommend acceptance in its current form.